# Effects of Fludioxonil on the Cell Growth and Apoptosis in T and B Lymphocytes

**DOI:** 10.3390/biom9090500

**Published:** 2019-09-18

**Authors:** Gun-Hwi Lee, Kyung-A Hwang, Kyung-Chul Choi

**Affiliations:** Laboratory of Biochemistry and Immunology, College of Veterinary Medicine, Chungbuk National University, Cheongju, Chungbuk 28644, Korea; rha131@naver.com (G.-H.L.); hka9400@naver.com (K.-A.H.)

**Keywords:** fludioxonil, lymphocyte, apoptosis, cell cycle arrest

## Abstract

Fludioxonil is fungicide used in agriculture, which is present in fruits and vegetables. In this study, the effects of fludioxonil on human immune cell viability, apoptosis, cell cycle arrest, and mitochondrial membrane potential were examined in human immune cells, such as Jurkat T cells and Ramos B cells. To examine the cell viability, Jurkat T cells and Ramos B cells were treated with fludioxonil (10^−9^–10^−5^ M) for 24 h and 48 h. Water soluble tetrazolium salt assay showed that fludioxonil decreased Jurkat T cell and Ramos B cell viability. Jurkat T cell viability decreased at 24 and 48 h, but Ramos B cell viability decreased only at 48 h. JC-1 dye revealed decreased mitochondrial membrane potential in fludioxonil-treated Jurkat T cells and Ramos B cells. To evaluate apoptosis, annexin-V conjugated FITC, AF488, and propidium iodide (PI) were used and to evaluate cell cycle arrest PI was used. Apoptosis and cell cycle arrest were induced by fludioxonil (10^−7^–10^−5^ M) in the Jurkat T cells at 24 and 48 h and Ramos B cells at 48 h. Moreover, the protein levels of pro-apoptotic proteins, such as p53, BAX, and cleaved caspase 3, were increased and anti-apoptotic protein Bcl-2 was decreased by fludioxonil. Expression of the Fas receptor related to the extrinsic apoptosis pathway was increased by fludioxonil. Additionally, cyclin D1 and cyclin E1 were decreased by fludioxonil. In the present study, fludioxonil induced immunotoxicity in human T cells and B cells through apoptosis and cell cycle arrest. Therefore, the present study suggests that fludioxonil induces the cellular toxicity in immune cells.

## 1. Introduction

Pesticides are chemical substances that kill living fungi, weeds, insects, and rodents. Though pesticides benefit human life via increasing agricultural productivity through protecting crops from fungi, weeds, and insects, etc., pesticides are toxic to humans and theenvironment. Many pesticides have been used in agriculture, such as organochlorines (OCs), organophosphorus (OPs), carbamate (CBs), pyrethroids, chlorophenoxys, triazines amid, and phthalimides. These pesticides remain in the air, water, and soil [1,2]. According to human health issues on pesticides released by the Environmental Protection Agency (EPA) of the US, they present that residual pesticide exposure occurs through the skin, eyes, inhalation, and ingestion. Residual pesticides, exposed to humans, are a hazard to human health, and can lead to disorders in nerves, dysfunction of the immune system, and endocrine disruption [3,4]. Additionally, pesticides are known to be harmful in humans, but are frequently used in agriculture. 

Fludioxonil is a phenylpyroll fungicide and is an antifungal antibiotic, produced in 1993 by Sygenta [5]. Fludioxonil is used for cereals, fruits, and vegetables. It has a broad spectrum of activity [6]. Additionally, fludioxnil inhibits transport-associated phosphorylation of glucose that decreases mycelial growth [7]. Fludioxonil showed endocrine disruptor activity in androgen receptors in engineered human breast cancer cells, and induced genotoxicity through DNA damage in human cell lines, such as HepG2, ACHN, SY5Y, and LS-174T cells [8,9]. These studies imply that it may have deleterious effects on the human body as well as on fungi.

The immune system plays the role of the defense system from a variety of agents, such as pathogens, viruses, bacteria, toxins, and foreign materials. The immune system is classified into innate and adaptive immune systems, and the adaptive immune system is classified into humoral immunity and cellular immunity [10]. The innate immune system is induced when damaged, injured, or stressed cells send signals to the immune cells, such as dendritic cells, macrophages, monocytes, neutrophils, and epithelial cells. Innate immune systems are non-specific and dominant systems of host defense in most organisms [11,12]. The adaptive immune system developed in vertebrates and acts on stronger immune responses. The adaptive immune system is antigen-specific and requires the recognition of non-self antigens during a process called antigen presentation [13,14]. The cells of the adaptive immune system include T cells and B cells. T cells and B cells are major types of lymphocytes and are derived from hematopoietic stem cells in the bone marrow. B cells are implied in the humoral immune response; on the other hand, T cells are implied in the cell-mediated immune response [15].

Cell cycle is a mechanism that replicates itself and induces cell proliferation. This mechanism is proceeded by various proteins, such as cyclin and cyclin-dependent kinases (Cdks). Therefore, deregulation of cell cycle-related protein induces cell cycle arrest, causing apoptosis [16,17]

Apoptosis was originally introduced by Kerr in 1972 as a form of programmed cell death [18]. Biochemistry leads to cell changes, such as morphology, blebbing, cell shrinkage, nuclear fragmentation, and chromosomal DNA fragmentation [19]. Apoptosis is divided into two pathways. One is the extrinsic pathway—this pathway occurs through death receptors, such as TNF and Fas receptors. The receptor binds to a ligand, then the apoptosis signaling delivers to the cytoplasm and activates caspase [20,21]. The other pathway is the intrinsic pathway. This pathway is activated by the mitochondria. When the cell is damaged by DNA damage and oxidative stress, then apoptosis is induced through the mitochondria [22,23]. In the end, two pathways activate caspase 3. As a result, activated caspase 3 induces apoptosis [24]. Additionally, lost membrane potential in mitochondria can cause apoptosis, because mitochondria with depolarized membrane potential do not produce adenosine triphosphate (ATP) [25].

In this study, the cytotoxicity of fludioxonil in human immune cells, such as T cells and B cells, was investigated. To investigate the cytotoxicity of fludioxonil, cell proliferation, cell cycle arrest, apoptosis, mitochondrial membrane potential, and apoptosis- and cell cycle-related gene expression were evaluated.

## 2. Materials and Methods 

### 2.1. Reagents and Chemicals

Fludioxonil was purchased from Sigma-Aldrich Corp. (St. Louis, MO, USA). Fludioxonil was dissolved in 100% dimethyl sulfoxide (DMSO; Sigma-Aldrich Corp.). The final concentration of DMSO in the cell culture media was 0.1%.

### 2.2. Cell Cultures

The human Jurkat T cell line, human Ramos B cell line, and human lung diploid fibroblast cell line, WI-26, were obtained from the Korean Cell Line Bank (KCLB) (Seoul, Republic of Korea). Jurkat T cells and Ramos B cells were maintained in Roswell Park Memorial Institute (RPMI) 1640 medium supplemented with 10% fetal bovine serum (FBS), 10 U/mL penicillin, 100 µg/mL streptomycin, and 10 mM HEPES. WI-26 cells were maintained in Dulbecco’s Modified Eagle’s medium (DMEM; HyClone Laboratories Inc., Logan, UT, USA) added to 10% FBS, 10 U/mL penicillin, 100 µg/mL streptomycin, and 10 mM HEPES.

### 2.3. Treatment of Fludioxonil

The human Jurkat T cells and Ramos B cells were seeded at a density of 6 × 10^5^ cells/mL in the 24-well plates (SPL Life Science, Pocheon, Republic of Korea) and 10^7^ cells/mL in 90 mm cell culture dish (SPL Life Science). The seeded cells in 24-well plate were used to analyze apoptosis, cell cycle arrest, and mitochondrial membrane potential, and seeded cells in 90 mm dish were used for the extraction of proteins for Western blot analysis. Then, T cells and B cells were treated by fludioxonil at 10^−7^–10^−5^ M for 24 or 48 h and incubated at 37 °C in 5% CO_2_.

### 2.4. Water Soluble Tetrazolium Salt (WST) Assay

The human Jurkat T cells, Ramos B cells, and WI-26 cells were seeded at a density of 5 × 10^4^ cells/well in 200 µL in 96 well plates (SPL Life Science), and incubated in a humidified 5% CO_2_ atmosphere at 37 °C for 24 h. Incubated cells were treated with various concentrations of fludioxonil (final concentration 10^−9^–10^−5^ M in the medium) for 24 or 48 h. After 24 and 48 h, the cell viability was evaluated using EZ-Cytox (Daeil Lab, Seoul, Republic of Korea). EZ-Cytox 20 µL was added to each well and incubated at 37 °C for 1 h. The absorbance was measured at 450 nm using a microplate reader (BioTek Instruments, Winooski, VT, USA). 

### 2.5. FACS Analysis of Cell Cycle Arrest by PI Staining

In the 24-well plate, Jurkat T cells and Ramos B cells treated with fludioxonil (10^−7^ M–10^−5^ M) for 24 and 48 h were collected to 15 mL tubes, and collected cells were centrifuged at 400× *g* for 10 min and removed the supernatant. The cells were washed by phosphate-buffered saline (PBS) and resuspended by PBS, then the Jurkat T cells and Ramos B cells were fixed by 70% ethanol and the cells were incubated in 4 °C for 1 h. After cells were fixed, these cells were washed by PBS and stained with propidium iodide (PI) premix, which combined PI (Sigma-Aldrich Corp.) and RNase A (iNtRON Biotechnology, Seongnam, Republic of Korea). Then, cells, treated with PI, incubated at 4 °C for overnight. These cells, stained with PI and fixed with ethanol, were analyzed by flow cytometry (SH-800; Sony Biotechnology Inc., Bothell, WA, USA) to evaluate cell cycle arrest. A total of 30,000 cells were analyzed by flow cytometry.

### 2.6. FACS Analysis of Apoptosis by FITC, AF488-Annexin V/PI Staining

In the 24-well plate, Jurkat T cells and Ramos B cells were treated with fludioxonil (10^−7^ M to 10^−5^ M) for 24 and 48 h. The treated Jurkat T cells and Ramos B cells were collected to 15 mL tubes, and stained with FITC, Alexa Fluor 488-annexin V/PI (Invitrogen, Carlsbad, CA, USA) for 30 min. After staining, Jurkat T cells and Ramos B cells were washed by PBS. Then the stained Jurkat T cells and Ramos B cells were counted and 30,000 cells were analyzed by flow cytometry. Early apoptotic cells were defined as positive annexin V/negative PI and late apoptotic cells were defined as positive annexin V/positive PI.

### 2.7. Analysis of Mitochondrial Membrane Potential

In the 24-well plate, the immune cells were treated with fludioxonil (10^–7^ M–10^–5^ M) for 24 and 48 h and collected each 15 mL tubes, centrifuged at 400× *g* for 10 min, and the supernatant was removed. Remaining cells in the 15 mL tubes were stained by JC-1 dye (Invitrogen, Carlsbad, CA, USA) 10 µg/mL at 37 °C in 5% CO_2_ incubator for 15–30 min. The stained cells were washed by PBS twice and resuspended in media. The stained cells were seeded in 0.2% gelatin-coated 24-well plates, and stained cells were incubated at 37 °C in 5% CO_2_ overnight. The stained cells were observed with an Olympus CKX 41 microscope (Olympus Corp., Tokyo, Japan) under green fluorescence and red fluorescence. The red fluorescence indicates healthy mitochondria and the green fluorescence indicate depolarized membrane potentials of mitochondria.

### 2.8. Western Blot Assay

After treatment with fludioxonil in 10^−7^ M–10^−5^ M cells, proteins from the Jurkat T cells and Ramos B cells were extracted by radioimmunoprecipitation assay (RIPA) buffer (50 mM Tris-HCl (pH 8.0), 150 mM NaCl, 1% NP-40, 0.5% deoxycholic acid, and 0.1% SDS). The protein concentrations were determined by a solution of bicinchoninic acid (BCA; Sigma-Aldrich Corp.) which was mixed with copper II sulfate solution (Sigma-Aldrich Corp.) at a ratio of 50:1. The total cell proteins (30 µg) were separated in a 15% SDS-PAGE gel and transferred to a polyvinylidene fluoride (PVDF) membrane (BioRad Laboratories, Hercules, CA, USA). The membrane was immersed in primary antibodies in tris-buffered saline (TBS) containing 5% bovine serum albumin (BSA; RMBIO Corp., Missoula, MT, USA), as shown in Table 1, and incubated overnight at 4 °C. The membrane was incubated with horseradish peroxidase (HRP)-conjugated secondary antibody (goat anti-mouse IgG or goat anti-rabbit IgG; BioRad Laboratories) for 2 h. proteins were detected by LuminoGraph II (ATTO, Tokyo, Japan) using EZwestLumi plus (ATTO).

### 2.9. Reverse Transcription-Polymerase Chain Reaction (RT-PCR)

Total RNA was isolated from human Jurkat T cells and Ramos B cells using TRIzol Reagent (Invitrogen Life Technologies, Waltham, MA, USA). After isolating RNA, cDNA was synthesized using moloney murine leukemia virus reverse transcriptase (M-MLV RT) (iNtRON Biotechnology) from the isolated mRNA. Fas gene was amplified from cDNAs of Jurkat T cells and Ramos B cells to confirm gene expression levels. In this analysis, to compare *Fas* gene expression, the human glyceraldehydes-3-phosphate dehydrogenase (GAPDH) was used as an internal control as shown Table 2. To analyze the PCR product on 1.5% agarose gel containing NEOgreen (NEO Science, Suwon, Republic of Korea), electrophoresis was used. The bands representing the target genes were detected by LuminoGraph II (ATTO).

## 3. Results

### 3.1. Fludioxonil Exposure Supressed Cell Viability

The effects of fludioxonil on the viability of Jurkat T cells and Ramos B cells were measured by WST assay. A decrease in cell viability was observed after exposure of Jurkat T cells to fludioxonil for 24 and 48 h. In Jurkat T cells treated with fludioxonil for 24 h, cell viability was decreased by only a high concentration (10^−5^ M). Additionally, in Jurkat T cells treated with fludioxonil for 48 h, cell viability was decreased by dose dependent in 10^−8^–10^−5^ M (Figure 1A). However, in the Ramos B cells treated with fludioxonil for 24 h, cell viability did not change, and in Ramos B cells treated with fludioxonil for 48 h, cell viability was decreased by of fludioxonil at concentrations of 10^−7^–10^−5^ M in a dose-dependent manner (Figure 1B). In WI-26 cells, which are human lung fibroblasts, treated with fludioxonil (10^−9^–10^−5^ M) for 24 and 48 h, cell viability did not change (Figure 1C).

### 3.2. Fludioxonil Exposure Induced Cell Cycle Arrest

Flow cytometry was used to confirm the changes in cell cycle induced in Jurkat T cells and Ramos B cells that were control or exposed to fludioxonil (10^−7^–10^−5^ M) for 24 or 48 h. To confirm the effects on the cell cycle by fludioxonil, Jurkat T cells and Ramos B cells were stained by PI. Separation of cells in G0/G1, S phase, and G2/M was based on the linear fluorescence intensity after PI. In Jurkat T cells treated with fludioxonil (10^−7^–10^−5^ M) for 24 or 48 h, the G0/G1 phase was increased by fludioxonil in a concentration-dependent manner. However, the S phase was decreased by fludioxonil (Figure 2A). At Ramos B cells which exposure of fludioxonil for 24 or 48 h, the S phase did not change with fludioxonil, but the G0/G1 phase was decreased by exposure of fludioxonil in a concentration-dependent manner (Figure 2B). 

### 3.3. Alteration of Cell Cycle-Related Genes Expression by Fludioxonil

To evaluate the effects of fludioxonil on the protein expression of cell cycle-related genes, such as cyclin D1 and cyclin E1, a Western blot assay was used using proteins which extracted from Jurkat T cells and Ramos B cells treated with fludioxonil (10^−7^–10^−5^ M) for 24 or 48 h. In Jurkat T cells, protein expression of cyclin D1 and cyclin E1 was decreased by exposure to fludioxonil (10^−7^–10^−5^ M) at 24 and 48 h compared to control, but there was significance at the highest concentration of fludioxonil (Figure 3A). In Ramos B cells treated with fludioxonil (10^−5^ M) for 48 h, protein expression of cyclin D1 was decreased, and cyclin E1 was decreased with exposure to fludioxonil (10^−7^–10^−5^ M) at 24 and 48 h (Figure 3B).

### 3.4. Apoptosis of Immune Cells by Fludioxonil Exposure

To confirm the effect of fludioxonil on apoptosis, Jurkat T cells and Ramos B cells following the treatment of fludioxonil were measured by FITC or AF488-annexin V/PI staining. Annexin V^+^/PI^–^ and Annexin V^+^/PI^+^ sections show apoptosis. At 24 and 48 h, apoptosis of Jurkat T cells was increased by fludioxonil (10^−7^–10^−5^ M) compared to control, but significance was present only at a high concentration of fludioxonil (10^−5^ M) (Figure 4A). In the Ramos B cells, apoptosis did not significantly increase at 24 h. However, at 48 h, apoptosis of Ramos B cells was increased by fludioxonil (10^−7^–10^−5^ M), but significance was presented at only at a high concentration of fludioxonil (10^−5^ M) (Figure 4B). Apoptosis was increased in a time- and concentration-dependent manner in Jurkat T cells and Ramos B cells at 48 h, but at 24 h apoptosis was increased by fludioxonil in only Jurkat T cells. Thus, it was confirmed that fludioxonil can induce apoptosis in the lymphocytes, and Jurkat T cells are more sensitive as fludioxonil than Ramos B cells.

### 3.5. Decrease of Mitochondrial Membrane Potential by Fludioxonil

Fluorescence microscopy was used to assess the dysfunction of mitochondrial membrane potential in Jurkat T cells and Ramos B cells, as shown Figure 5. Fludioxonil treatment resulted in a considerable increase in green fluorescence which represented a decrease in the mitochondrial membrane potential. In both Jurkat T cells and Ramos B cells treated with fludioxonil for 24 or 48 h, green fluorescence was increased (Figure 5). At Jurkat T cells treated with fludioxonil for 24 h, green fluorescence presented at fludioxonil 10^−6^ M, and at treatment of fludioxonil for 48 h in Jurkat T cells, green fluorescence presented at fludioxonil 10^−7^ M. Additionally, the green fluorescence was increased by fludioxonil in a concentration-dependent manner (Figure 5A). In the Ramos B cells treated with fludioxonil for 24 h, green fluorescence appeared strongly at 10^−5^ M fludioxonil. Unlike the treatment of fludioxonil for 24 h, in the treatment of fludioxonil for 48 h, green fluorescence was present in 10^−7^ M fludioxonil. In addition, the green fluorescence increase by fludioxonil was concentration-dependent (Figure 5B). Thus, these results show that fludioxonil induced apoptosis by destroying the mitochondrial membrane potential.

### 3.6. Alteration of Apoptosis-Related Genes Expression by Fludioxonil

Western blot was accomplished to investigate whether the fludioxonil affects the protein expression of apoptosis-related genes, such as p53, Bcl-2-associated X protein (BAX), Bcl-2, and cleaved caspase-3, in Jurkat T cells and Ramos B cells or not (Figure 6). The pro-apoptotic protein expression of p53, BAX, and cleaved caspase-3 were increased, but the anti-apoptotic protein expression of Bcl-2 was decreased by fludioxonil in Jurkat T cells treated with fludioxonil for 24 and 48 h. However, the quantified results present that cleaved caspase-3 and BAX was significantly increased, and Bcl-2 was significantly decreased (Figure 6A). In the Ramos B cells treated with fludioxonil for 24 and 48 h, the protein expression of p53, BAX, and cleaved caspase-3 was increased and protein expression of Bcl-2 was decreased, at 24 h protein expression of cleaved caspase-3 was presented only in 10^−5^ M fludioxonil. In the Ramos B cells treated with fludioxonil for 24 h, significance was presented in p53, cleaved caspase-3, and Bcl-2, and in the Ramos B cells treated with fludioxonil for 48 h, significance was presented in cleaved caspase-3 and BAX (Figure 6B).

### 3.7. Increase of Fas Gene Expression by Fludioxonil

Fas is related with the extrinsic apoptotic pathway. Semi-quantitative RT-PCR was used to confirm the gene expression of *Fas* by the treatment of fludioxonil in Jurkat T cells and Ramos B cells (Figure 7). In Jurkat T cells treated with fludioxonil for 24 or 48 h, expression of *Fas* was increased by fludioxonil. Especially at 48 h, expression of *Fas* was more increased than the control (Figure 7A), but in the Ramos B cells, fludioxonil did not alter the expression of the *Fas* gene (Figure 7B).

## 4. Discussion

Pesticides used for agricultural products increase the quality and protection from pests, insects, fungi, and harmful weeds. Pesticides have properties of residue, and pesticide residues are easily exposed to the environment and tend to be environmentally persistent [2]. These pesticide residues cause many diseases and health risks, such as cancer, disrupting the nervous system, allergies, and immune system disorders [3,4]. According to another study, atrazine, carbamate, and tributylin were demonstrated to induce apoptosis, cell cycle arrest, and dysfunction in immune cells by affecting the expression of apoptosis-related genes and cell cycle-related genes, and by decreasing mitochondrial membrane potential [28,29,30,31]. In particular, fludioxonil was investigated in regard to promoting oncogenesis [8,27,32]. Although fludioxonil has shown to induce oncogenesis, there was no research in which fludioxonil affects the immune system. In particular, we employed Jurkat T cells and Ramos B cells because we considered that T cells and B cells were basic immune cells in adaptive immunity. Jurkat T cells and Ramos B cells are immortalized lines of human lymphocytes used to study cell signaling pathways. Jurkat T cells have the ability to produce interleukin 2, while Ramos B cells have the characteristics of B lymphocytes and are used as a model of B lymphocytes in apoptotic studies. In this study the effects of fludioxonil exposure on apoptosis and cell proliferation of human Jurkat T cells and human Ramos B cells was investigated by evaluating the apoptosis- and cell cycle-related genes, and the dysfunction of mitochondrial membrane potential. 

First, the effects of fludioxonil on human immune cells, such as Jurkat T cells and Ramos B cells were identified and investigated. To investigate the cell viability, fludioxonil concentrations (10^−9^ M to 10^−5^ M) were employed for cell viability assay. In human immune cells, fludioxonil decreased the Jurkat T cells’ and Ramos B cells’ viability. In the Jurkat T cells, cell viability was decreased in 24 and 48 h. On the other hand, Ramos B cells’ viability was decreased in only 48 h. The cell viabilities in Jurkat T cells and Ramos B cells were decreased at 48 h by fludioxonil (10^−7^–10^−5^ M). Thus, in the other experiments, these concentrations of fludioxonil (10^−7^–10^−5^ M) were used to decrease the cell viability of Jurkat T cells and Ramos B cells. Decreases in cell viability mean that fludioxonil has an effect on cell proliferation. Decreased immune cell proliferation showed that immune cell counts diminished, thus, decreases in immune cell counts may induce immunodeficiency. T cells and B cells are known to play a role in adaptive immunity, are major types of immune cells [33], and are derived from hematopoietic stem cells, but have distinguished roles in immune response. T cells are involved in cellular immunity and B cells play a role in the humoral immunity. T cells activate B cells, macrophages, and cytotoxic T cells, and kill the infected cells and tumor cells. B cells secrete the antibodies that eliminate the antigens [34,35]. Furthermore, fludioxonil did not affect the cell viability of normal cells, such as the human lung diploid fibroblast cell line WI-26. These results suggest that fludioxonil decreased the cell viability in immune cells but did not affect normal cells.

In addition, fludioxonil induced cell cycle arrest. Cell cycle arrest by fludioxonil was confirmed using flow cytometry by PI staining. Then, Western blot assay proved that fludioxonil modulated the protein expression of cell cycle-related genes, such as cyclin E1 and cyclin D1. These proteins are related with cell cycle G1 and S phase and critical kinases in cell cycle regulation [36]. Cyclin D1 applies in the G1 phase and cyclin E1 plays a role in the cell cycle G1/S transition [37,38]. In Jurkat T cells, the G0/G1 phase was increased and the S phase was decreased by fludioxonil, and protein expression of cyclin E1 and cyclin D1 was decreased by fludioxonil at 24 and 48 h. In the Ramos B cells, fludioxonil decreased the G0/G1 phase and decreased the protein expression of cyclin D1 at 24 and 48 h, but the protein expression of cyclin E1 did not change at 24 h. The decrease of cyclin E1 implies that the G0/G1 phase does not progress to the S phase, and the decrease of cyclin D1 implies that the G0/G1 phase was decreased. In Jurkat T cells, G1 cell cycle arrest was induced by fludioxonil [39]. In cell cycle analysis, the change at 24 h in Ramos B cells was not linked to the expression of cyclins. However, both cyclin E1 and cyclin D1 were decreased by fludioxonil in Jukat T cells. Therefore, fludioxonil has effects on the cell cycle via alteration of cell cycle regulation proteins, such as cyclins. Cell cycle arrest was induced by fludioxonil in human Jurkat T cells and Ramos B cells.

The apoptosis by fludioxonil in Jurkat T cells and Ramos B cells were confirmed by decreasing mitochondrial membrane potential, analysis of annexin V/PI staining, and altered protein expression of apoptosis-related genes and increased gene expression of *Fas/Fas* ligand. In Jurkat T cells, early apoptosis (Annexin V^+^/PI^-^), late apoptosis (Annexin V^+^/PI^+^), and green fluorescence related with mitochondrial membrane potential loss were increased by treatment with fludioxonil for 24 and 48 h. In Ramos B cells, early apoptosis, late apoptosis, and green fluorescence were increased by treatment with fludioxonil for 48 h. Annexin V is phospholipid-binding protein that binds to phosphatidylserine (PS). PS is exposed at the cell surface which underwent apoptosis and recognized the apoptotic cells [40]. The dysfunction of mitochondrial membrane potential leads to apoptosis. The mitochondrial membrane potential was measured by JC-1, and JC-1 dye exhibits two fluorescences: one is red fluorescence, reflecting higher membrane potential, and the other is green fluorescence, indicating dysfunction of membrane potential [41,42]. Proteins related with apoptosis were confirmed by Western blot assay. p53, BAX, and cleaved caspase-3 were increased in Jurkat T cells that were treated with fludioxonil for 24 and 48 h, but Bcl-2 was decreased in Jurkat T cells that were exposed to fludioxonil for 24 and 48 h. In Ramos B cells, protein expression of p53, BAX, Bcl-2, and cleaved caspase-3 was altered with the exposure to fludioxonil for 24 and 48 h. p53 can induce the apoptotic pathway in the cell, and p53 up-regulated BAX, which is a pro-apoptotic protein, and down-regulated Bcl-2, which is an anti-apoptotic protein [43,44]. These proteins activated caspase-3 to cleaved caspase-3, and cleaved caspase-3 induces apoptosis [45]. These proteins are related to the intrinsic apoptotic pathway, and the intrinsic apoptotic pathway occurs through the mitochondria [46]. The other apoptosis pathway is the extrinsic apoptotic pathway, which is related with the Fas/Fas ligand. Fas and Fas ligand induce apoptosis through activating caspase-3 [47]. *Fas* gene expression was confirmed. In Jurkat T cells, *Fas* was increased by fludioxonil treatment for 24 and 48 h. However, in Ramos B cells treated with fludioxonil for 24 and 48 h, fludioxonil did not change Fas. As a result, fludioxonil induces apoptosis via the increase in pro-apoptotic-related proteins, *Fas* gene, a decrease in anti-apoptotic-related protein, and induction of dysfunction of mitochondrial membrane potential. Thus, these results present that fludioxonil may induce intrinsic and extrinsic apoptotic pathways and dysfunction of mitochondrial membrane potential in human immune cells, Jurkat T cells, and Ramos B cells.

## 5. Conclusions

In conclusion, it was found that fludioxonil may induce apoptosis by inhibiting cell proliferation through cell cycle arrest. Apoptosis was induced by the intrinsic apoptotic pathway mediated in mitochondria and the extrinsic apoptotic pathway and dysfunction of mitochondrial membrane potential. In addition, Jurkat T cells appeared to be more sensitive than Ramos B cells in this study. Therefore, as shown in Figure 8, the present study indicates that fludioxonil pesticides can induce cytotoxicity in human immune cells. That is, fludioxonil might cause of various immunological diseases by cytotoxicity.

## Figures and Tables

**Figure 1 biomolecules-09-00500-f001:**
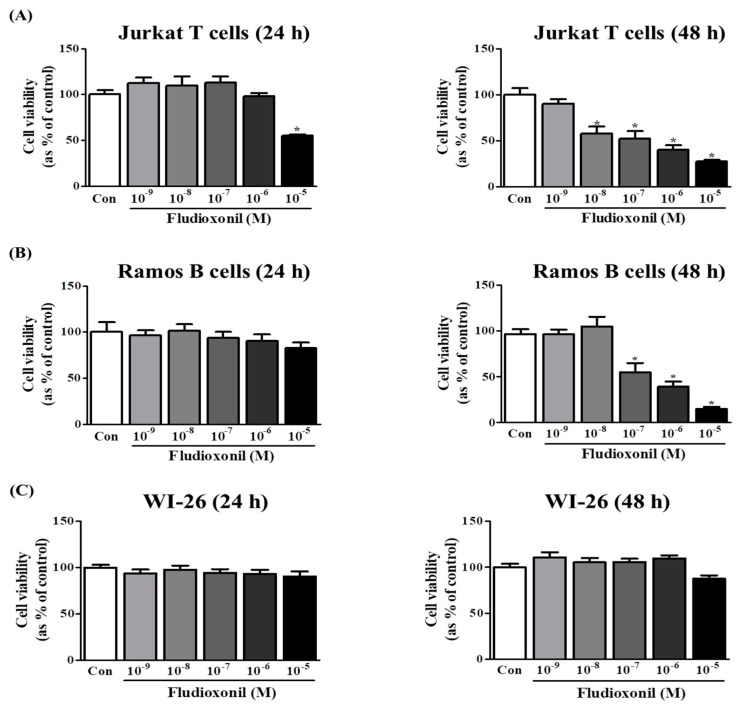
Viability of Jurkat T cells and Ramos B cells following the treatment of fludioxonil. Jurkat T cells, Ramos B cells and WI-26 cells (50,000 cells/well) were seeded in 96-well plates and treated with 10^−9^–10^−5^ M fludioxonil for 24 or 48 h. Cell viability was confirmed by WST assay. DMSO was the control vehicle. (**A**) Jurkat T cells, (**B**) Ramos B cells, and (**C**) WI-26 cells. The data are shown as the means ± standard deviation (SD) of three independent experiment. **p* < 0.05 as compared with the control.

**Figure 2 biomolecules-09-00500-f002:**
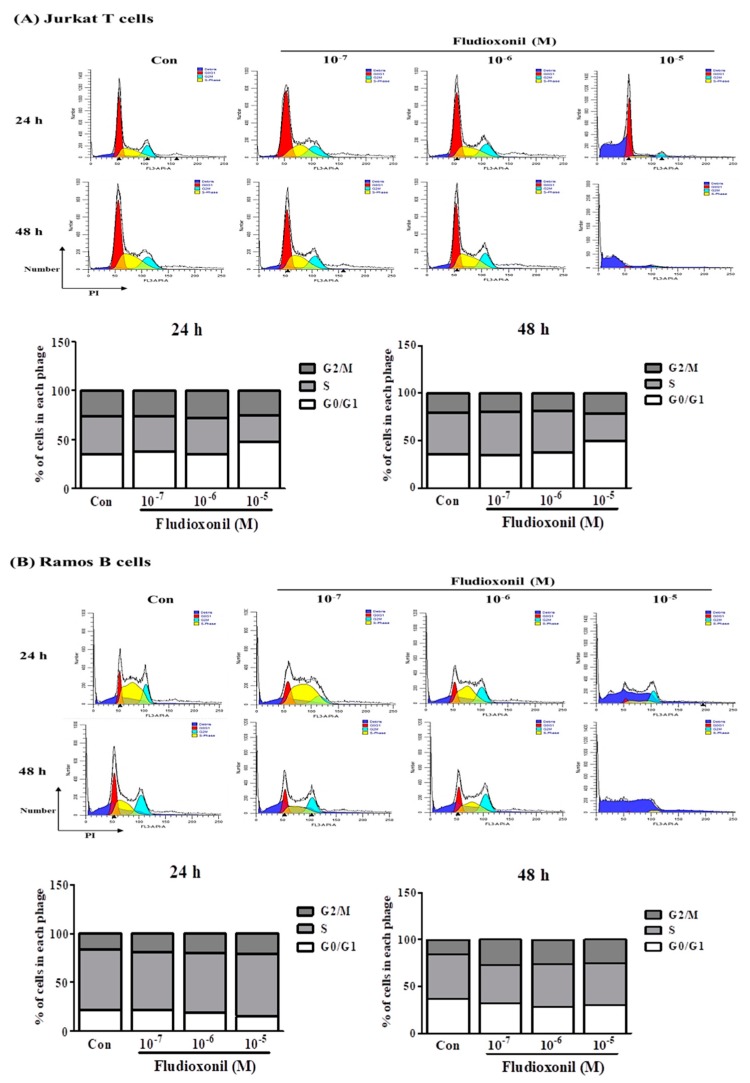
Cell cycle arrest of Jurkat T cells and Ramos B cells at the treatment of fludioxonil. (**A**) Jurkat T cells and (**B**) Ramos B cells histogram graph were representative cell cycle analysis results using flow cytometry. The cell cycle was analyzed by cell staining with PI. In the histogram results, the dark blue section indicates the debris of the cells, the red section indicates G0/G1 phase, the yellow section indicates the S phase, and light blue indicates the G2/M phase. FL3-A represents the intensity of PI and the y-axis represents the cell numbers. In the stacked bar graphs, the changes of G0/G1, S, and G2/M phase proportion of Jurkat T cells and Ramos B cells show the altered cell cycle by fludioxonil.

**Figure 3 biomolecules-09-00500-f003:**
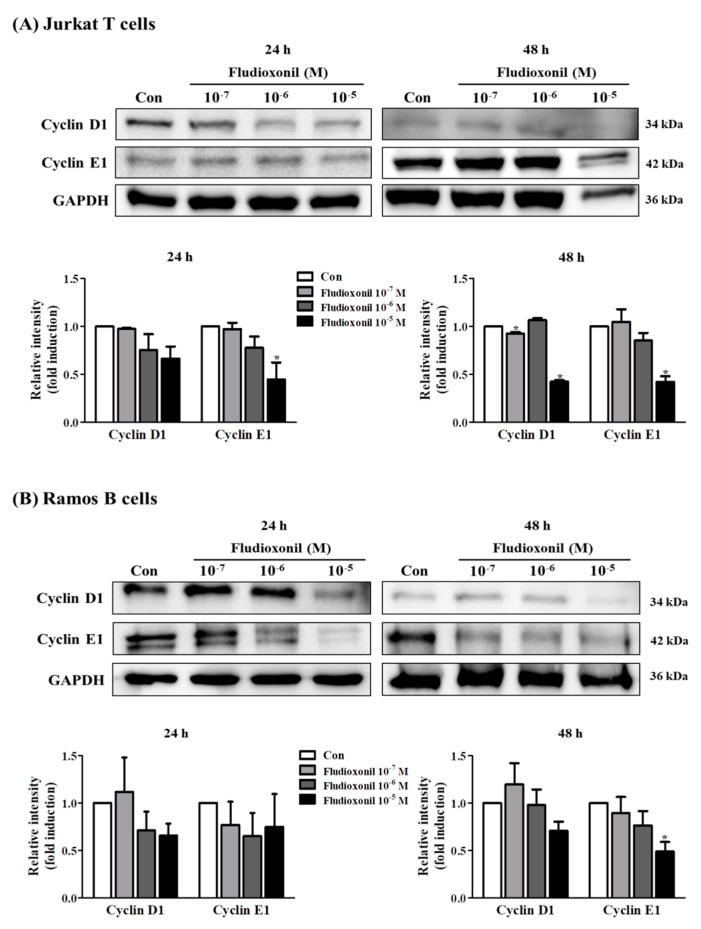
Protein expression of cell cycle-related genes in Jurkat T cells and Ramos B cells by treatment of fludioxonil. (**A**) Jurkat T cells and (**B**) Ramos B cells were treated with 10^−7^–10^−5^ M fludioxonil for 24 or 48 h. The cell lysates were prepared and the target proteins were analyzed by Western blotting. The protein expression of cell cycle-related genes, such as cyclin E1 and cyclin D1, in protein levels are shown as representative Western blot images. All bands were normalized by GAPDH and presented as % of controls. The value of the control (DMSO) was set as 1 and the values of the treatment groups were indicated as fold induction vs. control. The data are shown as the means ± standard deviation (SD) of three independent experiment. **p* < 0.05 as compared with the control.

**Figure 4 biomolecules-09-00500-f004:**
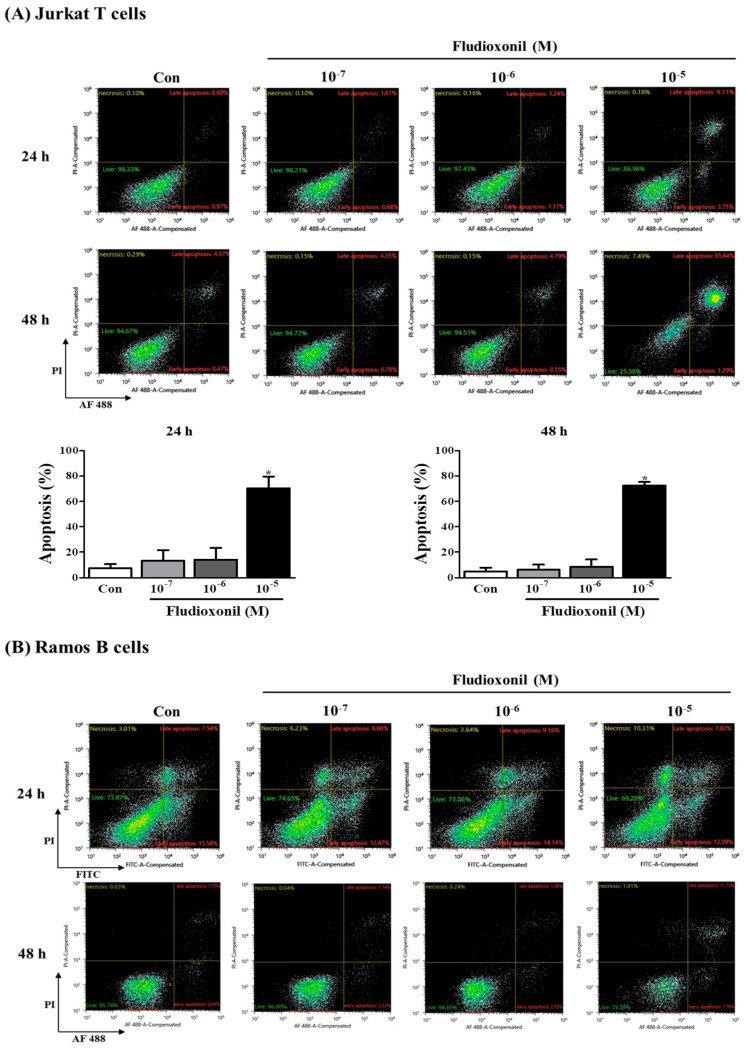
Apoptosis of Jurkat T cells and Ramos B cells by fludioxonil. (**A**,**B**) The results of flow cytometry analysis of annexin V and PI staining in Jurkat T cells and Ramos B cells treated with fludioxonil (10^−7^–10^−5^ M) for 24 or 48 h are shown. The x axis is annexin V and the y axis is PI. (**A**) Jurkat T cells (**B**) Ramos B cells. The live cell populations are in the lower-left quadrant (Annexin V^–^/PI^–^), the cells in early apoptosis are in the lower-right quadrant (Annexin V^+^/PI^–^), and late apoptosis is shown in the upper-right quadrant (Annexin V^+^/PI^+^). Statistical graphs of annexin V/PI staining are shown, and data present the apoptosis cell ratio from three independent experiments. Apoptotic cells include annexin V^+^/PI^–^ cells and annexin V^+^/PI^+^ cells. The data are shown as the means ± standard deviation (SD) of three independent experiment. **p* < 0.05 as compared with the control.

**Figure 5 biomolecules-09-00500-f005:**
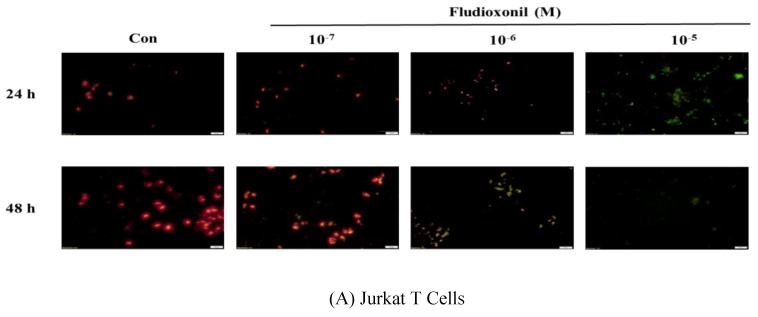
Detection of mitochondrial membrane potential by JC-1 staining after fludioxonil treatment. Decrease of aggregate red fluorescence and increase of green fluorescence indicate a loss in mitochondrial membrane potential. (**A**) Jurkat T cells and (**B**) Ramos B cells treated with 10^−7^–10^−5^ M fludioxonil for 24 and 48 h. Each panel is representative of three separate experiments, and these results are merged with green and red fluorescence. Scale bar = 20 μm.

**Figure 6 biomolecules-09-00500-f006:**
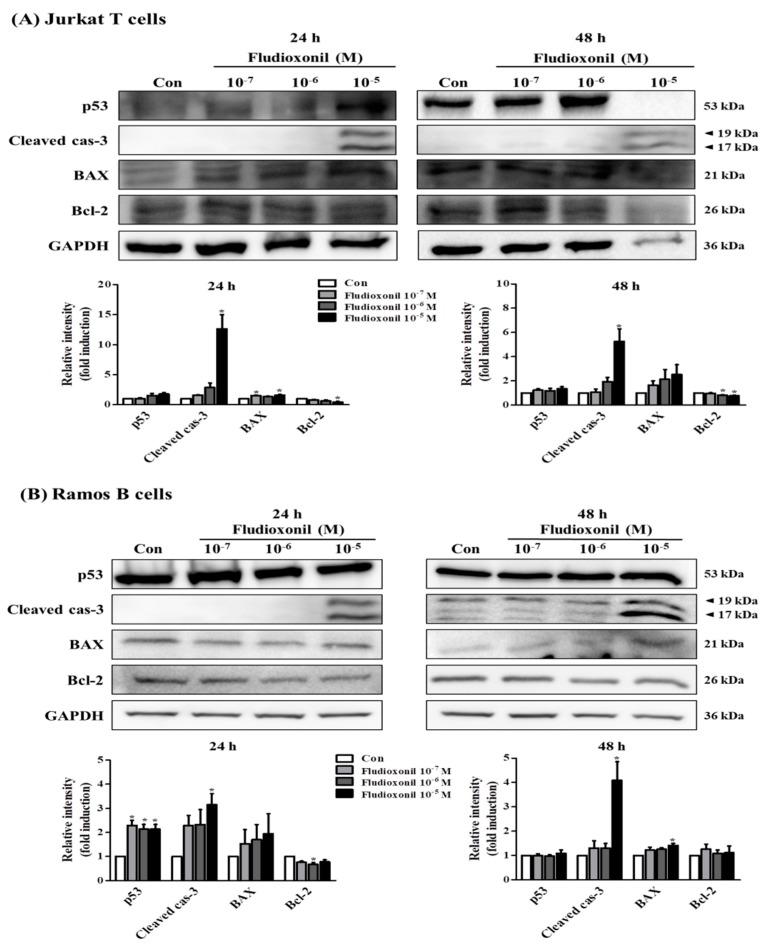
Protein expression of apoptosis-related genes in Jurkat T cells and Ramos B cells. (**A**) Jurkat T cells and (**B**) Ramos B cells were treated with fludioxonil 10^−7^–10^−5^ M for 24 or 48 h. The cell lysates were prepared and the target proteins were analyzed by Western blotting. The protein expression of apoptosis-related genes, such as p53, BAX, cleaved caspase-3, and Bcl-2, are shown as representative Western blot images. All bands were normalized by GAPDH and presented as % of controls. The value of the control (DMSO) was set as 1 and the values of the treatment groups were indicated as fold induction vs. control. The data are shown as the means ± standard deviation (SD) of three independent experiment. **p* < 0.05 as compared with the control.

**Figure 7 biomolecules-09-00500-f007:**
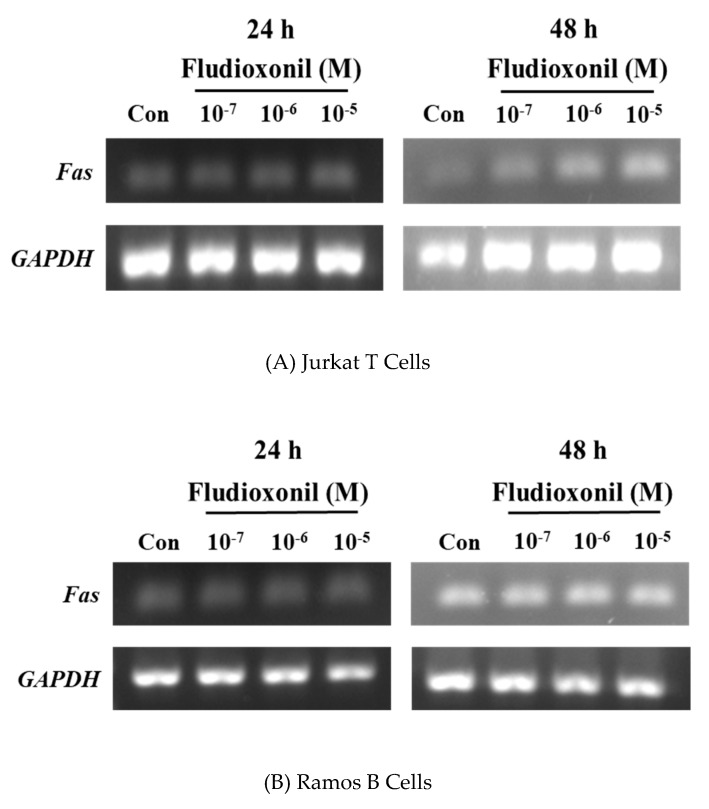
Expression of the *Fas* gene in Jurkat T cells and Ramos B cells. Jurkat T cells and Ramos B cells (100,000 cells/well) were seeded in 24-well plates, respectively, and treated by fludioxonil (10^−7^–10^−5^ M) for 24 or 48 h. Total RNA was extracted by TRIzol Reagent in Jurkat T cells and Ramos B cells. Total RNA converts to cDNA. Then, *Fas* gene expression was confirmed by semi-quantitative RT-PCR. (**A**) In Jurkat T cells, fludioxonil (10^−7^–10^−5^ M) treated for 24 or 48 h. (**B**) Ramos B cells were treated with fludioxonil (10^−7^–10^−5^ M) for 24 or 48 h. Each panel is representative of three separate experiments.

**Figure 8 biomolecules-09-00500-f008:**
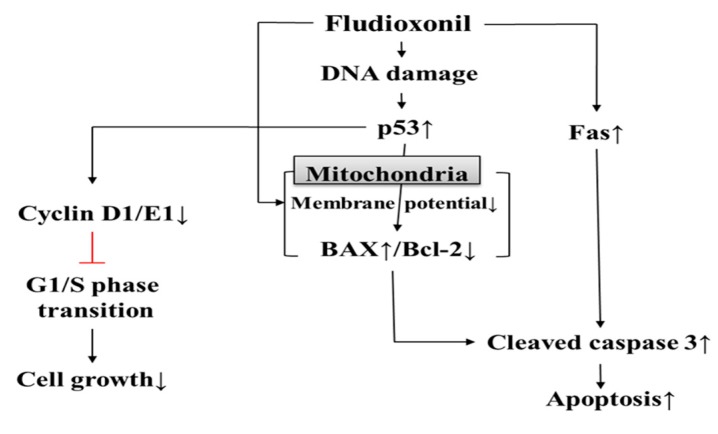
Cytotoxicity of fludioxonil on human immune cells. Fludioxonil leads to apoptosis that is mediated by the mitochondria and death receptor (*Fas*). In mitochondria, fludioxonil depolarizes membrane potential and increases expression of p53, then p53 alters the expression of BAX and Bcl-2. Increased BAX activates cleaved caspase-3. Fludioxonil increases gene expression of *Fas,* which activates cleaved caspase-3. Activated cleaved caspase-3 induces apoptosis. Additionally, fludioxonil decreases the expression of cell cycle-related genes, such as cyclin D1 and E1, and decreased cyclin induces cell cycle arrest.

**Table 1 biomolecules-09-00500-t001:** Name and source of antibodies used in this study.

Proteins	Company	Cat No.	Description	Western Blot
p53	Santa Cruz	SC-126	Mouse mAb	1:1000
BAX	Online	ABIN 135027	Mouse mAb	1:1000
Bcl-2	Bio Legend	658702	Mouse mAB	1:500
Cleaved caspase-3	Cell signaling	D175	Rabbit pAb	1:2000
Cyclin D1	Abcam	Ab187364	Mouse mAb	1:2000
Cyclin E1	Abcam	Ab3927	Mouse mAb	1:2000
GAPDH	Abcam	Ab8245	Mouse mAb	1:10000

**Table 2 biomolecules-09-00500-t002:** Oligonucleotide sequences of the semi-quantitative RT-PCR products.

Target Genes	Primer Sequence (5′-3′)		Product Size (Bp)	Reference
*Fas*	F	**5**′-TGAAG GACAT GGCTT AGAAG TG-3′	118	[26]
	R	**5**′-GGTGC AAGGG TCACA GTGTT-3′		
*GAPDH*	F	**5**′-ATGTT CGTCA TGGGT GTGAA CCA-3′	351	[27]
	R	**5**′-TGGCA GGTTT TTCTA GACGG CAG-3′		

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
