# Peer review of "Effects of Fludioxonil on the Cell Growth and Apoptosis in T and B Lymphocytes"

_biomolecules, 2019, doi:10.3390/biom9090500_

Round 1
Reviewer 1 Report
Fludioxonil is antifungal fungicide, this work describes the effects of fludioxonil on human immune cells viability, apoptosis, cell cycle arrest and mitochondrial membrane potential using Jurkat T cells and Ramos B cells for the first time. Demonstrating fludioxonil pesticides can induce cytotoxicity in human immune cells which may cause immunological disease. This draft is well organized and this reviewer suggests its publication in present form.
Revision suggestions.
1. Systematic English checking and corrections is needed. eg. P5, line 178, “concentration-dependentmanner”; P6, line 194, “…..was decreased…”---“…decreased….”, check the whole text and do corrections; P6, line 196 “In Ramos B cells which treatment fludioxonil…”---“In Ramos B cells which were treated by fludioxonil…”; P9, line 244, “…..apoptosis by decreased mitochondrial….”—“….apoptosis by destructing mitochondrial….”
2. P6, line 198, delete “But there was no significance at treatment for 24 h (Figure 3B).”,this saying is not true when the concentration of fludioxonil reached 10-5 M.
3. Figure 2, the resolution of flow cytometry diagrams should be improved and edited by using original electronic data, to make them suitable for publication.
Reviewer 2 Report
The paper entitled: “Effects of fludioxonil on the cell growth and 2 apoptosis in human T and B lymphocytes” presents some new data concerning the effects of a fungicide (fludioxonil) on cells viability, apoptosis, cell cycle arrest and mitochondrial membrane potential. Even the paper presents a study of interest concerning the possible effect of the human exposure to fungicides on immune system, the paper is very confused and some times hard to read, due to the extensive errors in English language and Grammar.
General Comments
It seems that the paper wanted to show that the fungicide affects rather immune cells system than other type of cells. This information is very poorly presented even that it could be any interesting point of departure. WI-26 cell line, appear from time to time, but the role assessment of the cell proliferation on this type of cell can be hardly deducted from the paper. Is this type of cells representative for “normal cells”? Why only a line of lung fibroblast was tested and not another cell type? Why they are different than immune cells? The proliferation rate is lower? How were these cellular types choose instead of other?
Concerning the experimental design and the cellular model.
We suppose that Jurkat T cell/Ramos B cells were used in this study as models for cellular/humoral mediated immunity. This aspect could be only deducted from the context and need more explanations. Also, the choice of these two cell models should be commented. The discussion should be oriented versus the differences observed between the two type of cells and should provide and explanation about the different findings related to the different role that these cells have in the accomplishment of the immune response. For example, there were differences in the cell cycle progression between the two cell models that were linked in the discussion section by the authors with differences in the cyclins. However, similar effects of the fungicide were observed on cyclines that could not explain the differences observed for the cell cycle arrest.
Again, the choice of the concentrations of fludioxonil (10-5 to 10-7 M) for the assay other than cell proliferation should be explained.
The present discussion is in principal a repetition of the results. An interpretation of the results and a comparation with the results of other researchers should appear instead.
Other comments
M&M section
Line 87. Treatment with fludioxonil. It is hard to understand the purpose of seeding cells in 24 wells plates and in culture dishes, as no other explanation was provided. It will be better to specify the culture conditions (type of plate, the number of seeded cells etc) for each type of analyse
Line 93. The are differences between the no of seeded cells in the text and in the Fig 1.
Line 101, Line 127. The concentration of fludioxonil used for the different assays should be indicated
Line 105 What means premix?
Table 1. GAPDH should be included
Round 2
Reviewer 2 Report
The authors have answered to all my comments and requirements. The article could be accepted in its present form after a revision of the English language.